# A Designed Host Defense Peptide for the Topical Treatment of MRSA-Infected Diabetic Wounds

**DOI:** 10.3390/ijms24032143

**Published:** 2023-01-21

**Authors:** Alex Vargas, Gustavo Garcia, Kathryn Rivara, Kathryn Woodburn, Louis Edward Clemens, Scott I. Simon

**Affiliations:** 1Department of Biomedical Engineering, University of California, Davis, CA 92037, USA; 2Riptide Bioscience, Inc., Vallejo, CA 94592, USA; 3Department of Dermatology, University of California, Davis, CA 95618, USA

**Keywords:** antimicrobial peptides (AMPs), designed Host Defense Peptide (dHDP), MRSA, foot ulcers, TALLYHO, diabetes, neutrophils, wound healing

## Abstract

Diabetes mellitus is a chronic disease characterized by metabolic dysregulation which is frequently associated with diabetic foot ulcers that result from a severely compromised innate immune system. The high levels of blood glucose characteristic of diabetes cause an increase in circulating inflammatory mediators, which accelerate cellular senescence and dampen antimicrobial activity within dermal tissue. In diabetic wounds, bacteria and fungi proliferate in a protective biofilm forming a structure that a compromised host defense system cannot easily penetrate, often resulting in chronic infections that require antimicrobial intervention to promote the healing process. The designed host defense peptide (dHDP) RP557 is a synthesized peptide whose sequence has been derived from naturally occurring antimicrobial peptides (AMPs) that provide the first line of defense against invading pathogens. AMPs possess an amphipathic α-helix or β-sheet structure and a net positive charge that enables them to incorporate into pathogen membranes and perturb the barrier function of Gram-positive and Gram-negative bacteria along with fungi. The capacity of skin to resist infections is largely dependent upon the activity of endogenous AMPs that provided the basis for the design and testing of RP557 for the resolution of wound infections. In the current study, the topical application of RP557 stopped bacterial growth in the biofilm of methicillin-resistant *Staphylococcus aureus* (MRSA) USA300 infected wounds on the flanks of clinically relevant diabetic TALLYHO mice. Topical application of RP557 reduced bacterial load and accelerated wound closure, while wound size in control diabetic mice continued to expand. These studies demonstrate that RP557 reduces or eliminates an infection in its biofilm and restores wound-healing capacity.

## 1. Introduction

Diabetes mellitus (DM) is a disease characterized by persistent hyperglycemia, insulin resistance, and chronic dysfunctional metabolism resulting in progression of destructive inflammatory cycles [1,2]. A disruption of homeostatic metabolic processes accelerates cellular senescence and dysregulation of normal immune functions that produces many chronic diabetic complications. Patients with diabetes typically have increased amounts of white adipose and visceral fat that are a major source of inflammatory cytokines, including TNF-α, IL-1β, and IL-6, along with chemokines that attract neutrophils and monocytes [3,4]. Contributing to diabetic wound chronicity is an environment that has increased systemic circulation of inflammatory cytokines, senescent cells that are unable to proliferate or mount an immunological defense and a milieu that encourages bacterial infections in biofilm [5]. Diabetic foot ulcers (DFUs) are thus a product of enhanced oxidative stress that impairs healing by disrupting the synchrony of healing, thereby prolonging acute inflammatory responses marked by the influx of myeloid and T-cells and hastening wound chronicity rather than resolution of infection [3]. Most diabetic wounds become polymicrobial infections, whereby a biofilm comprising symbiotic bacteria, yeast, and fungal loads [6] silently spread, invading deep tissue and bone, which can lead to lower-limb amputation. Developing therapies for DFUs is thus a major scientific challenge and unmet clinical need, as well as a growing socioeconomic burden that requires new approaches to enhance and synchronize the innate immune response in immunocompromised patients [7,8,9].

Designed host defense peptides (dHDPs) are synthetic analogs to naturally occurring antimicrobial peptides (AMPs) found in myeloid and skin resident cells [10]. These peptides have evolved over millennia to mount the first line of defense against invading pathogens, including Gram-positive, Gram-negative bacteria, and fungi [11,12,13,14]. Their amphipathic α-helix or β-sheet structure has a net positive charge that facilitates their insertion into plasma membranes, thus perturbing the barrier function of negatively charged bacterial walls, resulting in rapid bacterial killing even in antibiotic-resistant strains such as MRSA. The capacity of normal tissues to resist infections is dependent upon the activity of AMPs. It is in the diabetic wound where the AMPs of the innate immune system have been crippled that bacteria thrive [15]. Previous studies demonstrated that when applied to *P. aeruginosa*-infected mouse corneas, dHDPs significantly reduce bacterial burden and the inflammatory response [16]. These proof-of-concept studies led to the discovery and development of RP557 that is the result of three iterative design cycles of peptide synthesis and subsequent evaluation of its antimicrobial activity, cytotoxicity, and proteolytic susceptibility [17,18]. 

Our recent reports demonstrated that the polymicrobial biofilm infection of *S. aureus* and *P. aeruginosa* in porcine burn wounds was resolved with a single 2% dose of the dHDP RP557 [19]. In the current study, we evaluated RP557 as a therapeutic agent for the treatment of DFU by assessing its capacity to augment the innate immune response in the clearance of a bacterial infection and to hasten wound resolution in diabetic mice. We found that RP557 is synergistic with neutrophil antibacterial activity in effectively reducing bacterial load and promoting wound closure in a mouse model of type 2 diabetes. 

## 2. Results

An early response of the innate immune system to a cutaneous infected wound is neutrophil degranulation that mediates changes in adhesion receptor expression and function and elicits the release of proteases and reactive oxygen species. This accompanies recruitment of neutrophils to the site of tissue insult where they release proteases and produce reactive oxygen species (ROS), as well as phagocytose bacteria, which marks the inflammatory phase of infection [20]. We first determined the dose–response effect of RP557 on human neutrophil function since they respond similarly to mouse neutrophils in response to an *S. aureus* infection and reagents for detecting activation as a function of conversion of β_2_-integrin to a high affinity state are readily available. Neutrophil viability, degranulation, activation of β_2_-integrins, and migratory capacity were assayed. Neutrophils remained 90% viable at an RP557 dose of 312 μg/mL, and decreased to 60% at 600 μg/mL (Figure 1a). At concentrations that maintained high viability, RP557 upregulated surface CD11b nearly three-fold and activated high-affinity CD18 that supports neutrophil migration and phagocytosis nearly six-fold (Figure 1b,c). L-selectin is maximally expressed on circulating mature neutrophils and is shed upon activation with chemotactic stimulation by proteolytic cleavage [21]. Treatment with RP557 at 78 μg/ml elicited ~40% shedding of L-selectin (CD62L), whereas 312 μg/mL elicited cleavage to undetectable levels of receptor expression (Figure 2a). Transmigration of neutrophils across porous transwell membranes was maximally stimulated by IL-8 at 10 nM and in a dose-dependent manner by RP557 (Figure 2b). Reactive oxygen species’ production was activated by RP557 alone at 156 μg/mL and increased at 312 μg/mL to a level higher than that stimulated by IL-8. Stimulation with fMLP, a potent activator of ROS production, increased levels three-fold from baseline. Co-stimulation of neutrophils with RP557 amplified ROS production in a dose-dependent manner and was more potent than IL-8 (Figure 2c). Taken together, these data indicate that RP557 functions as a chemotactic factor and pro-inflammatory agonist for neutrophils and enhances their antibacterial functions including integrin function, migration, and production of ROS.

We next examined the capacity of RP557 to attract neutrophils to the site of a sterile full-thickness wound in C57BL/6 mice that stably express the EGFP-lysozyme M (lysM) gene upregulated in mature neutrophils [22,23]. Daily treatment with RP557 to uninfected nondiabetic wounds had no effect on increasing the basal level of neutrophil migration within wounds nor did it influence the rate of wound closure over time (Figure 3a,b). This indicates that RP557 was not cytotoxic in vivo and did not interfere with the normal inflammatory process and subsequent wound contraction and closure.

RP557 was next applied to diabetic mouse wounds infected with MRSA USA300 to assess its capacity to cooperate with the innate immune response and hasten bacterial clearance and wound resolution. Among the factors that lead to chronicity in DFUs is the presence of a biofilm in wounds [2]. We first assessed whether MRSA formed a biofilm 24 h post-inoculation using scanning electron microscopy, which confirmed characteristic formation of a biofilm within the wound bed (Figure 4a). To evaluate its effectiveness in killing MRSA biofilms in wounds, RP557 was dosed at 24 h post-wounding, and infection and bioluminescence signal were measured to quantify its effect on the dynamics of bacterial growth on subsequent days of treatment. At Day 3 following wounding and infection and 48 h following application of the first dose of RP557 (Figure 4b), bacterial growth was markedly slower in treated wounds than in the control group. The daily application of RP557 continued to significantly suppress bacterial growth, while the control group plateaued at a peak level over days 4–5 of infection. Over the ensuing 7–9 days, the innate immune system effectively resolved the infection in all mice. Images of MRSA bioluminescence confirmed diminishing viability of bacteria in wounds being treated with RP557 as compared to that of the vehicle control (Figure 4c). 

We next compared the effect of RP557 on wound healing in uninfected and infected wounds of TALLYHO diabetic mice. In this study, RP557 had no effect on wound closure in uninfected diabetic wounds (Figure 5a). In contrast, a therapeutic effect of RP557 in defeating an MRSA infection and hastening wound closure was detected within 24 h after application of the first treatment (Figure 5b). Within 48 h after the second dose of RP557, treated wounds were significantly smaller than vehicle-control wounds and continued to rapidly close with daily topical application. A comparison of the effect of RP557 on the rate of wound closure in infected versus uninfected wounds revealed that RP557 markedly increased the rate of closure of infected wounds and returned them to that of uninfected wounds (Figure 5c) within five days. Examination of the quality of wound resolution with RP557 treatment in representative images 24 h after receiving a fourth dose revealed increased desiccation and contraction accounting for its effect on wound closure (Figure 5d).

## 3. Discussion

Neutrophil recruitment to murine dermal wounds during the inflammatory stage is required for bacterial clearance but is also dispensable for healing in sterile wounds, as several studies, including ours, have shown that limiting their number in circulation can hasten wound closure [24]. Here we report that the designed host defense peptide RP557 had potent antibacterial activity, diminished the rate and extent of growth of an MRSA biofilm in obese diabetic murine wounds, and significantly accelerated the rate of wound closure. Analogous to human α-defensins that are stored in the primary granules of neutrophils and, when released into phagosomes or inflamed tissue, exert broad host-defense functions, RP557 elicited degranulation and activation of β_2_-integrins. It was chemotactic in stimulating migration at 156 μg/mL (~75 μM) and amplified ROS production at levels nearly equivalent to those of IL-8, a potent chemokine activator of neutrophils. The mechanism of action that accounts for the proinflammatory activity of RP557 was not addressed in these studies. However, treatment with RP557 did not perturb the normal process of neutrophil recruitment into sterile wounds of wild-type C57BL/6 mice, nor did it impede the rate of wound closure in these mice. Likewise, treatment of sterile wounds with RP557 in diabetic mice did not hasten the slower rate of wound closure compared to that in wild-type mice. In these uninfected mice, the rate of healing exhibited a linear trajectory (Figure 3b and Figure 5a), which is consistent with what others have previously reported [25,26]. 

Noteworthy is our finding that in the presence of an MRSA infection, RP557 dramatically reduced bacterial abundance and significantly hastened wound closure to a rate equivalent to that in uninfected wounds. Thus, RP557 functioned as a potent antibacterial peptide by directly activating neutrophil inflammatory functions in a manner that was synergistic with GPCR signaling to initiate wound healing. We have previously reported in a rodent model of vulvovaginal candidiasis that RP557 applied every 8 h in a 0.2% solution significantly reduced the fungal load. In this study, we observed effective fungicidal activity between 0.2% and 2% [17]. In a second study, we tested RP557 at a single dose of 0.2% in a nondiabetic mouse scratch wound model infected with MRSA and reported significant reduction of bacterial load within 24 h of application. In this model, a single dose suppressed bacterial growth for eight days of observation [10]. Here, we demonstrated in-vivo that RP557 did not inhibit wound closure at ~0.5 μM, and in vitro that neutrophils remained 90% viable at a dose of ~75 μM. Furthermore, neutrophil influx in LysM-EGFP C57Bl6 mice at a higher dose of 2% RP557 was not altered (Figure 3). Thus, we conclude that RP557 does not exhibit a narrow therapeutic window.

Diabetic wounds are exacerbated by a heightened inflammatory state in the presence of a weakened innate immune response, which is observed secondary to persistent hyperglycemia that can cause premature cellular senescence [27]. Resident cell replication is slowed, and a senescence-associated secretory phenotype is present in diabetic wounds that enhances inflammation, thereby amplifying the senescence of surrounding cells [28]. Diabetic-foot-ulcer infections are the manifestation of such a dysfunctional cell metabolic phenotype that perpetuates a destructive positive feedback cycle within injured tissue that presents a challenge to clinicians [1]. It is in this environment, often in aged thin skin of diabetics with reduced innate immune system function, that wounds are prone to infection by pathogens present on the skin surface. Following infection, a biofilm can rapidly form, which presents an added challenge that can shift homeostasis to chronicity of diabetic wounds [2]. Here, we demonstrate that topical treatment with RP557 penetrated the wound and rapidly neutralized an MRSA biofilm. While MRSA continued to proliferate in the vehicle control group, daily topical treatment with RP557 effectively inhibited bacterial growth over the duration of observation. The significant impact of bacterial killing on the healing process was further demonstrated by the fact that RP557 did not alter the rate of wound healing in wounds of uninfected TALLYHO mice, nor did it Influence neutrophil migration or wound closure in C57BL/6 mice. As noted in several reviews, successful treatment of diabetic foot ulcers involves coordinated efforts that address the various factors that perpetuate the comorbidities of this disease [2,7,29]. In addition to treatments for controlling plasma glucose levels, regular debridement and off-loading devices, direct care of wounds with multifunctional wound dressings results in increased healing rates [9,30]. Despite such interventions, MRSA wounds present a significant challenge due to virulence factors that prevent early neutrophil influx and promote biofilm formation [23]. Augmenting the antibacterial armamentarium by topical application of RP557 provides a means of controlling the early events during the inflammatory phase that promote a more rapid transition to wound closure.

MRSA sheds lipopolysaccharides from its membrane, and these can stimulate the generation of inflammatory cytokines, including TNF-α, IL-6, and IL-1β, to a greater extent in diabetic mice. The latter signaling may account for the greater inflammatory response than in infected nondiabetic mice [3]. Thus, it is critical to rapidly control bacterial growth and curtail the inflammatory phase of wound resolution. Although we did not measure neutrophil recruitment into the wounds of diabetic mice, it has previously been reported that it is delayed, as is their capacity to produce ROS in the presence of an *S. aureus* infection [31]. Despite impaired chemotactic capacity, neutrophil numbers continue to rise in diabetic wounds to compensate for the impaired bacterial killing and increase in growth [32]. We envision that RP557 affects resolution in at least two ways, by reducing the inflammatory stimulus through defeating MRSA growth and biofilm production and by augmenting neutrophil chemotactic and antibacterial functions, including phagocytosis and production of reactive oxygen species (ROS), within the wound bed.

Among the factors included in the direct treatment of infections is the topical application of antibiotics, superparamagnetic iron oxide nanoparticles, growth factors, follistatin-like protein-1, and other factors that have proven to exert positive effects that enhance the healing of diabetic wounds [33]. An advantage of RP557 as a topical antimicrobial versus application of a topical or systemic antibiotic is that neither *P. aeruginosa* nor MRSA are known to develop resistance to this dAMP [10]. This is an important consideration given the current rise in antibiotic resistance of MRSA and other bacteria and the fact that prolonged treatment required for healing diabetic wounds enhances the possibility of antibiotic-resistant infections. We conclude that, analogous to natural defensins, RP557 provides a unique approach in that it amplifies the intrinsic activity of chemotactic factors and may boost the innate immune response for improved treatment of diabetic foot ulcers when applied in conjunction with ancillary care that promotes regeneration of resident skin cells. 

## 4. Materials and Methods

### 4.1. Mice

Ten-week-old male diabetic TALLYHO/JNGJ mice were obtained from Jackson Laboratories (Stock No: 005314) and maintained on LabDiet 5K52 feed (Item 15502, Newco Distributors Inc., Hayward, CA, USA, MFG 5K52C3P) at the University of California, Davis. These mice were a model of type 2 diabetes as confirmed by measures of blood hyperglycemia (i.e., glucose level of >300 mg/dL). Additionally, LysM-EGFP–transgenic mice bred in animal facilities at the University of California were used. 

### 4.2. S. aureus Preparation

A bioluminescent strain of MRSA USA300 was cultured overnight on BHI agar plates with 5% cow blood (VMBS, UC Davis). The plate was scanned on an IVIS Spectrum (PerklinElmer, Waltham, MA, USA) to confirm bioluminescence. Single colonies were picked and placed in 6 mL of Tryptic Soy Broth medium with chloramphenicol (100 ug/mL) and kanamycin (50 ng/mL). The bacteria were cultured overnight in a shaking incubator (VWR) at 37 °C and 300 RPM. The next day, the bacteria were diluted in 6 mL of Tryptic Soy Broth medium with chloramphenicol (100 ug/mL) and cultured in a shaking incubator (VWR) at 37 °C and 300 RPM. The bacteria were cultured to an optical density of 0.0418 ± 0.0026, which corresponded to 1.14–1.29 × 10^7^ CFU in 50 uL (in the SEM study presented in Figure 4a, the final dose was 9.04 × 10^6^ CFU in 50 uL). Bacteria optical density (OD600) was measured using a NanoDrop spectrophotometer (ThermoFisher, Waltham, MA, USA). After that, the bacteria were centrifuged at 3850 RPM for 10 min at 4 °C (Beckman Coulter, Brea, CA, USA) and resuspended in 1 mL of ice-cold PBS (Gibco, Waltham, MA, USA). 

### 4.3. Mouse Wounding and Inoculation

Mice were wounded and inoculated as previously described [34]. Prior to wounding, the mice were given 0.03 mg/mL of buprenorphine hydrochloride (NDC 12496-0757-5CN) in a sodium chloride solution (NDC 63323-186-01) by an intraperitoneal injection. The mice were placed under 2% isoflurane (Vet One^®^ Fluriso™ NDC 13985-528-60) for 10 min, had their backs shaven and sterilized with gauze a in povidone-iodine solution (Cat # 3955-16, Ricca Chemical Company, Arlington, TX, USA) and gauze in 70% isopropanol, and received a full-thickness dorsal wound with a 6 mm biopsy punch. The wounds were subcutaneously inoculated with 50 uL of bacteria as prepared above or left uninfected. Once daily, wounds were photographed using a camera (Nikon, Tokyo, Japan) and scanned for bioluminescence using an IVIS Spectrum (PerklinElmer). Wound size was measured using ImageJ. Bioluminescence from the wounds of infected mice was measured by drawing a region of interest (ROI) on top of the wound using LivingImage^®^ Version 4.7.3 (Perklin Elmer, Waltham, MA, USA). Percent change was calculated relative to measurement at Day 1 post-infection, which was the first day of treatment.

### 4.4. RP557 Preparation and Application

The physiochemical properties of RP557 have been optimized to provide a relationship between net charge, amphipathicity and hydrophobicity that has resulted in decreased proteolytic degradation, broad spectrum antimicrobial (bacteria and fungi) and biofilm inhibition activity with reduced mammalian cytotoxicity, and an increased therapeutic index [10,11,17]. 



As shown above, RP557 is a 17 amino-acid peptide with two disulfide bonds. One bond is between the cysteines in the brackets indicating positions 3 and 16, and the second bracket denotes the cysteines in positions 7 and 12. These disulfide bonds thus form a hairpin-loop-like structure (see Figure 1 in reference [10]). RP557 was synthesized via solid phase synthesis (AmbioPharm, North Augusta, SC, USA). Peptide purity was >96% as assayed by high-performance liquid chromatography and mass spectroscopy [10]. A 20 mg/mL solution (2%) of peptide RP557 with a molecular weight of 2135 g/mol was prepared in PBS. Unless otherwise noted in the figure legend, the RP557 solution (50 uL) was topically applied onto dorsal TALLYHO mouse wounds once a day after imaging. As the 2% solution was 20 mg/mL, this meant that the 50 uL dose contained 1 mg of RP557. The first dose was administered on Day 1 post-infection. Mice were kept under anesthesia following imaging for 30–40 min to allow the peptide to absorb into the wound. 

### 4.5. Human Neutrophil Isolation

Whole blood was obtained from healthy donors who consented through the University of California, Davis institutional review board protocol #235586-9. Neutrophils were isolated from whole blood via negative enrichment using a direct human neutrophil isolation kit (EasySep™, STEMCELL Technologies, Cambridge, MA, USA). Briefly, 4 mL of whole blood was added to a 14 mL round bottom polystyrene tube, and then 200 µL of isolation cocktail and magnetic RapidSpheres™ each were added and incubated for 5 min. The tubes were topped up to 12 mL with PBS + 1 mM EDTA, and the cells were placed in an EasyEights™ magnet for 10 min. The top 10 mL of the cell solution were pipetted into a new tube and treated for an additional 5 min with 200 µL of magnetic RapidSpheres™ and placed on an EasyEights™ magnet for 5 min two times, each time collecting the entire cell solution. The cells were then spun down and resuspended in HBSS containing 0.1% HSA and kept at 1 × 10^7^ cells/mL on ice prior to use. Cell purity was >90%, determined using a Beckman Coulter Cell Counter. 

### 4.6. Quantification of Cell Viability

Neutrophils (1 × 10^6^ cells/ml) in an HBSS buffer containing Ca^2+^ and Mg^2+^ at 1 mM were incubated with RP557 for 30 min at 37 °C before washing with PBS twice and staining with 500 ng/mL propidium iodide for a cell viability dye. The cells were analyzed using an Attune NxT flow cytometer (ThermoFisher). The neutrophils were gated by their characteristic forward scatter vs. side scatter profile. 

### 4.7. Transwell Migration Assay

Neutrophil migration was quantified using a transwell chamber (Costar, Cambridge, MA, USA) with 24 mm diameter polycarbonate inserts (3 μm pore size). In the lower chamber, 2.6 mL of a medium containing varying concentrations of RP557, IL-8, or vehicle control was loaded. Then, 1.5 mL of neutrophils (1 × 10^6^ cells/mL) in an HBSS buffer containing Ca^2+^ and Mg^2+^ at 1 mM was added to the top of the polycarbonate transwell inserts. The plates were incubated at 37 °C and 5% CO_2_ for 2 h. The migrated cells were trypsinized and counted using a hemocytometer.

### 4.8. Flow Cytometry Detection of Cell Surface Marker Expression 

Neutrophils (1 × 10^6^ cells/mL) in an HBSS buffer containing Ca^2+^ and Mg^2+^ at 1 mM were treated with IL-8 or RP557 for 20 min at 37 °C before washing two times with a buffer and staining with antibodies (1.5–5 µg/mL, BioLegend, San Diego, CA, USA). The cells were stained with antibodies for high affinity (HA) CD18 (mAb24), L-Selectin (Dreg-56), and CD11b (M1/70) for 15 min at 37 °C before fixation with 1% paraformaldehyde at room temperature. 

### 4.9. Measurement of Reactive Oxygen Species (ROS) 

Neutrophils (1 × 10^6^ cells/mL) in an HBSS buffer containing Ca^2+^ and Mg^2+^ at 1 mM were incubated with the reactive oxygen species indicator Dihydrorhodamine 123 (2 µM) and treated with, RP557, IL-8, or vehicle for 10 min at 37 °C before addition of 1 µM N-formyl-Met-Leu-Phe (fMLP). The cells were incubated with N-formyl-methionyl-leucyl-phenylalanine (fMLP) fMLP at 37 °C for 5 min and then placed on ice to end reactions. Reactive oxygen species were then quantified through flow cytometry.

### 4.10. Statistical Analysis

Data analysis was performed using GraphPad Prism version 9.4.1 (GraphPad, San Diego, CA, USA). All data are presented as the mean ± standard error (SEM) unless otherwise specified. When appropriate, differences between matched pairs of conditions were analyzed for significant difference of the mean values by Student’s *t*-test. Significant differences between multiple groups were analyzed by performing multiple unpaired parametric *t*-tests using the linear step-up procedure of Benjamin, Krieger, and Yekutieli. *p* < 0.05 was considered significant. Statistically significant *p*-values are reported in each figure legend, respectively.

## Figures and Tables

**Figure 1 ijms-24-02143-f001:**
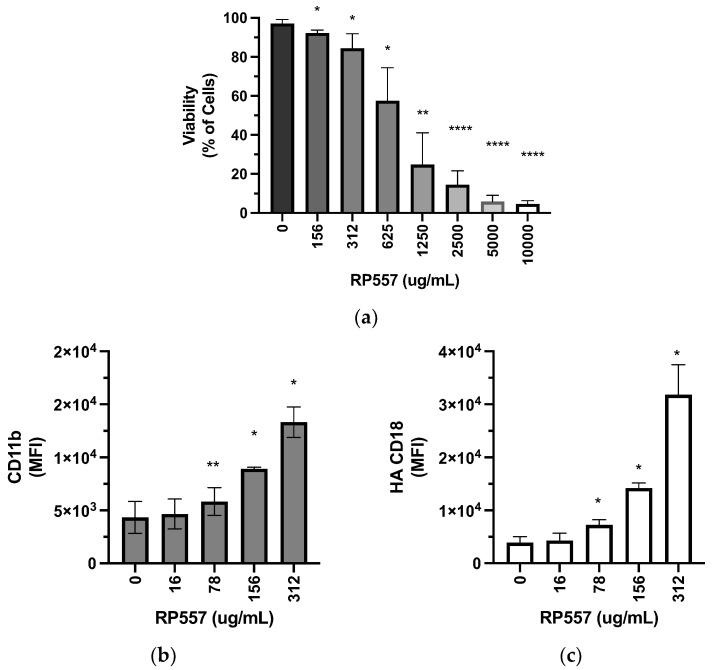
Neutrophil viability and integrin expression in response to incubation with RP557. (**a**) Viability of isolated human neutrophils incubated for 30 min with varying concentrations of RP557 solution as determined using propidium iodine fluorescence staining and flow cytometry. Paired ratio *t*-test was performed comparing the average value for an experimental condition to the untreated condition of the same donor (*n* = 4 per group). Isolated human neutrophils were incubated with varying concentrations of RP557 for 20 min at 37 °C, washed, and stained with antibodies for 15 min at 37 °C, followed by fixation and quantification by flow cytometry of (**b**) CD11b and (**c**) high affinity CD18. The data are presented as the mean fluorescence intensity ± SEM (*n* = 3 donors) with experimental replicates averaged for each donor. Paired *t*-tests were performed comparing the average value of the experimental condition to the 0 μg/mL control condition of the same donor. MFI: mean fluorescence intensity. *, **, and **** denote *p*-values of ≤0.05, ≤0.01, and ≤0.0001, respectively.

**Figure 2 ijms-24-02143-f002:**
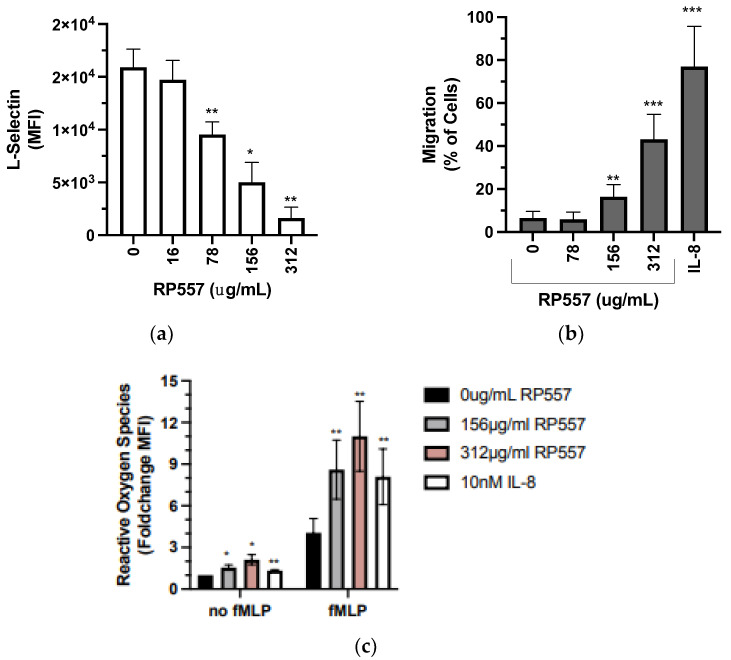
RP557 altered neutrophil effector function. (**a**) Isolated human neutrophils were incubated with varying concentrations of RP557 for 20 min at 37 °C, washed, and stained with antibodies for 15 min at 37 °C, followed by fixation and quantification by flow cytometry of CD62L expression on the cell surface. The data are presented as the mean fluorescence intensity ± SEM (*n* = 3 donors) with experimental replicates averaged for each donor. Paired *t*-tests were performed comparing the average value of the experimental condition to the 0 μg/mL control condition of the same donor. (**b**) Percent of neutrophils that migrated through an uncoated polycarbonate transwell insert in response to varying concentrations of RP557, Il-8 (10 nM), or vehicle control over a 2 h period (*n* = 3–5 per group). (**c**) Fold change in fluorescence over 0 μg/mL in neutrophils stained with 2 µM of the reactive oxygen species indicator DHR 123 and treated with RP557, IL-8, or vehicle control for 10 min before addition of 1 µM fMLP or vehicle control for 5 min. A paired ratio *t*-test was performed comparing the average value for an experimental condition to the untreated condition of the same donor. RP557 has a molecular weight of 2135.69 amu. MFI: mean fluorescence intensity. *, **, *** denote *p*-value ≤0.05 ≤0.01, and ≤0.001, respectively.

**Figure 3 ijms-24-02143-f003:**
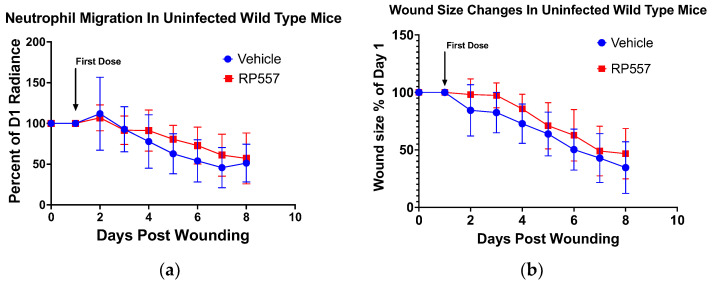
RP557 showed no toxicity in vivo. Nondiabetic LysM-EGFP mice that produced eGFP fluorescent neutrophils were wounded and treated with vehicle control or RP557. (**a**) Neutrophil recruitment into the uninfected wound was measured as a function of eGFP fluorescence radiance (photons/second) using an IVIS Spectrum. (**b**) Wound closure was also measured in these mice. *n* = 11 from three independent experiments.

**Figure 4 ijms-24-02143-f004:**
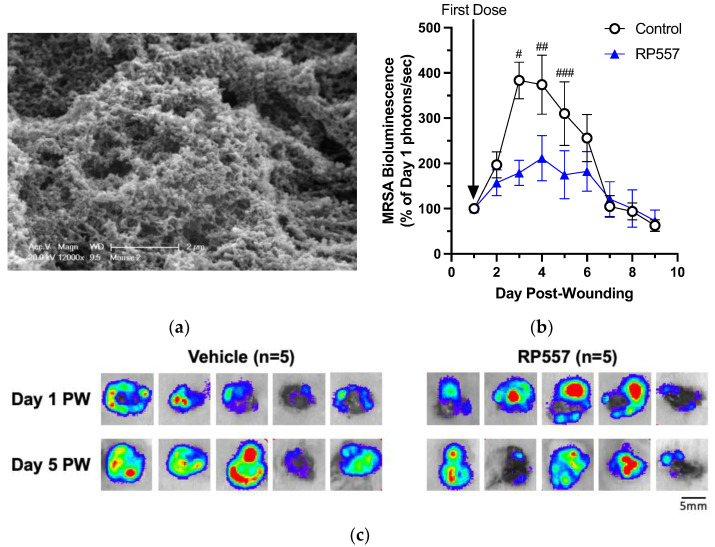
Effect of RP557 on bacteria killing against *S. aureus* in diabetic mouse wounds. TALLYHO diabetic mice were dorsally wounded and subdermally infected with bioluminescent MRSA USA300 (Day 0). Starting at one day after infection, mouse wounds were treated daily with either saline (control) or RP557. (**a**) SEM scan of the wound environment at 24 h post-infection showing that MRSA proliferated and formed a biofilm. (**b**) Wound bioluminescence was measured in an IVIS as photons/second and normalized to their value on Day 1. (**c**) Representative bioluminescence IVIS scans at one and five days after wounding. Data are from two independent experiments. *n* = 5–10 animals per group. #, ##, and ### denote *p*-values of <0.001, <0.070, and <0.150, respectively. Scale bar is 5mm.

**Figure 5 ijms-24-02143-f005:**
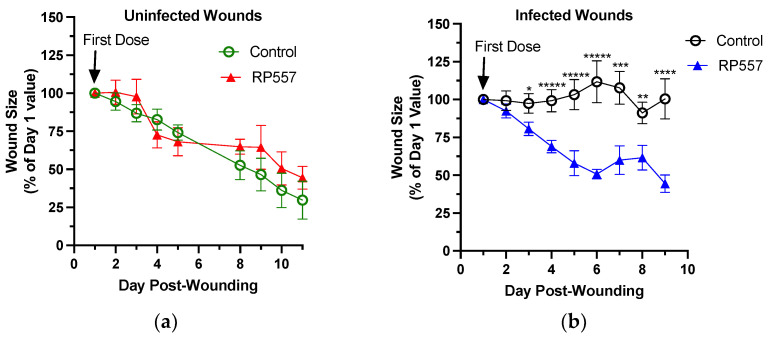
Effect of RP557 on healing of diabetic mouse wounds. TALLYHO diabetic mice were dorsally wounded on Day 0. Starting on Day 1, wounds were treated at each timepoint with control (saline or FBS) or RP557. (**a**) Wound closure of uninfected TALLYHO wounds as percent change relative to the size of the wound on Day 1. In this study, wounds were treated from Day 1–5 and Day 8–11. Data are from one experiment (*n* = 3–5 mice per group). (**b**) Wound closure of MRSA-infected TALLYHO wounds as percent change relative to the size of the wound on Day 1. Data are from two independent experiments (*n* = 5–10 animals per group). (**c**) Slope as a positive value of the simple linear regression line of the data shown in (**a**,**b**) from Day 1 to Day 5. (**d**) Representative images of the MRSA-infected wounds from (**b**) Day 1 and Day 5. *, **, ***, ****, and ***** denote *p*-values of <0.05, <0.03, <0.05, <0.005, and <0.003, respectively. Scale bar is 5mm.

## Data Availability

The data presented in this study are available on request from the corresponding author.

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
