# Peer review of "A Designed Host Defense Peptide for the Topical Treatment of MRSA-Infected Diabetic Wounds"

_ijms, 2023, doi:10.3390/ijms24032143_

Round 1

Reviewer 1 Report

“A Designed Host Defense Peptide for the Topical Treatment of MRSA Infected Diabetic Wounds”

This is an original research article in which the antimicrobial spectrum of RP557 is examined through a sensible battery of in vitro and in vivo tests. The peptide is synthesized on the bases of other known naturally occurring antimicrobial peptides that are detected in skin cells. The study is carefully done and targets one of the major clinical problems confronted on the clinical arena of diabetic foot wounds, and other types of chronic wounds. Importantly, the investigators have shown that the topical administration of their peptide reduced bacterial load and contribute to wound closure. This is an important study.

Please kindly read the following comments.

1-      I would suggest you to explain why or how non-infected wounds exhibit a linear healing trajectory (figure 5A) reminding wound closure pattern in healthy, non-diabetic animals - when the animals’ glycemia is declared to be above 300 mg/dl. This is fairly contrasting to what is shown in figure 5B.

2-      Representing wound closure kinetic on the bases of net calculated area is preferable that showing closure percentage referred to day 1 value. Please consider.

3-      Please consider to offer the data of living CFU contained in 50ul of bacteria suspension used to infect the wounds. Accordingly, provide the concentration of RP557 in the 50µl volume used to treat the wounds.

4-      Please consider to include in Discussion some comments on the dose/response profile of RP557. Isn’t it showing a narrow therapeutic window?

Author Response

Dear reviewer:

Thank you for reviewing our manuscript titled "A Designed Host Defense Peptide for the Topical Treatment of MRSA Infected Diabetic Wounds." We thank you for all your comments and suggestions. Please see the attachment below with our responses.

Best regards,

Alex Vargas

Reviewer 2 Report

My opinion about the article is positive, but I consider the authors should perform deep changes, mainly on the statistical analysis of the results and on the rearrangement of Materials and Methods section. 

Author Response

(The authors gave the same response as above.)

Reviewer 3 Report

The work is interesting. Diabetic wounds is an important area of research, the authors already reported the antibacterial activity in a previous paper this is an extension of that work. There are publication in the area and the use of antibacterial for wound healing is an approach. In this case the regeneration of the affected area is interesting.

The only observation that I have is to add the structure of RP557.  Add a figure with the structure of the compound since it is very specific.

Author Response

(The authors gave the same response as above.)
